# Persistent viral infections impact key biological traits in *Drosophila melanogaster*

Mauro Castelló-Sanjuán, Rubén González, Ottavia Romoli, Hervé Blanc, Jared C. Nigg¤*, Maria-Carla Saleh🆔*

Institut Pasteur, Viruses and RNA Interference Unit, Université Paris Cité, Paris, France

¤ Current address: Chan Zuckerberg Biohub, San Francisco, California, United States of America
* jared.nigg@czbiohub.org (JCN); carla.saleh@pasteur.fr (M-CS)

## Abstract

Persistent viral infections have been assumed to impose minimal fitness costs for insects. We established persistent mono-infections of *Drosophila melanogaster* with four different enteric RNA viruses: *Drosophila* A virus (DAV), *Drosophila* C virus (DCV), Bloomfield virus, and Nora virus. We observed that these infections significantly reduce fly survival, alter the number of viable offspring per female, modulate microbiome composition, impact locomotor abilities, and change activity patterns. These results demonstrate the significant impact of persistent viral infections on key biological traits and expand our understanding of the fitness costs of persistent viral infections for the host. In addition, the four viruses displayed different accumulation kinetics and elicited unique transcriptional profiles with no common core responses. The transcriptional changes triggered by DCV infection persisted even after viral clearance. This comprehensive comparative dataset represents a valuable resource for researchers studying host-pathogen interactions, providing detailed transcriptional profiles, and behavioral measurements across different viral infections and time points. Our findings reveal that persistent viral infections modulate critical aspects of insect biology, affecting host physiology and behavior.

## Introduction

The increasing availability of sequencing data has revealed tens of thousands of novel viruses in multiple host organisms, suggesting the widespread occurrence of previously undetected infections [1–3]. Recent studies have revealed extensive diversity in the RNA virosphere, identifying thousands of previously unknown viruses across diverse environments, including the most extreme habitats worldwide [4–6]. Among these newly discovered viruses are many that naturally infect *Drosophila* [3,7–9]. In some cases, these are persistent infections that represent a metastable state in which the physiological impacts of viral infection are maintained below a lethal threshold without leading to viral clearance [10,11]. In contrast to acute viral

**Data availability statement:** All data underlying the findings described in this manuscript are freely available. Individual numerical values supporting all summary data displayed in the figures and results are provided in Supporting Information files: underlying data for Fig 1 can be found in S1 Data, Fig 3 data in S2 Data, Fig 4 data in S3 Data, and S2 Fig in S4 Data. For Fig 2 (activity patterns and sleep behavior), the complete raw data files from *Drosophila* Activity Monitors along with clearly annotated code for replicating all analyses and generating plots are deposited in Zenodo at https://zenodo.org/records/17175344 (DOI: 10.5281/zenodo.17175344). RNA sequencing data for transcriptome analysis and bacteriome sequencing reads are publicly available in the NCBI Sequence Read Archive under BioProject accession number PRJNA1235228. All other supporting data, including viral quantification, survival analysis, offspring production, climbing assays, and bacterial load measurements, are included in the Supporting information files accompanying this article.

**Funding:** This work was supported by funding from the Agence Nationale de la Recherche (grant ANR-23-CE15-0038-01, INFINITESIMAL), the French Government's Investissement d'Avenir program, Laboratoire d'Excellence Integrative Biology of Emerging Infectious Diseases (grant ANR-10-LABX-62-IBEID), Fondation iXcore - iXlife - iXblue Pour La Recherche and the Explore Donation, MIE project to M.-C.S. R.G. salary is supported by a Pasteur-Roux-Cantarini fellowship of Institut Pasteur. This project has received funding from the European Union's Horizon 2020 research and innovation program under the Marie Skłodowska-Curie grant agreement No. 101024099 to J.C.N. The funders had no role in study design, data collection and analysis, decision to publish, or preparation of the manuscript.

**Competing interests:** The authors have declared that no competing interests exist.

**Abbreviations:** CLMM, cumulative link mixed models; CMV, cucumber mosaic virus; DAV, Drosophila A virus; DCV, Drosophila C virus; DEG, differentially expressed gene; dpe, days post-eclosion; GLM, generalized linear models; GLMM, generalized linear mixed models; HPV, human papillomavirus; OTUs, operational taxonomy units.

infections, which are characterized by a rapid process that results in the quick death of the host or in the clearance of the virus, persistent infections endure over extended periods within the individual host. During this time, the virus is still capable of being transmitted to other organisms, including the host's offspring, despite the inherent limitations imposed by the host's immune responses and cellular mortality [10–14].

Persistence is a frequent phenomenon in viral infections across all kingdoms, as it can provide advantages for both the virus and the host, acting as a modulator of the ecosystem. For instance, the cucumber mosaic virus (CMV) infects beet plants (*Beta vulgaris*), improving their drought tolerance and enhancing their freezing tolerance [15]. On the other hand, persistent human papillomavirus (HPV) infection must undermine host antiviral defense mechanisms, disrupt the balance of cellular proliferation and differentiation, and hijack DNA damage signaling and repair pathways to replicate viral DNA in stratified epithelium. Together, these modulations of host physiology carry detrimental consequences for the host, putting cells at high risk for carcinogenesis [16]. Nevertheless, research has mostly focused on the acute effects of viral infections. Consequently, the impacts of persistent infections on host physiology and behavior remain understudied.

In this study, we aimed to elucidate whether persistent infections in the fruit fly, *Drosophila melanogaster* have consequences on fly fitness or key biological traits. The fruit fly has an extensively characterized genome and short generation times, making it an ideal model organism for studying persistent viral infections throughout life span. Growing evidence shows that persistent infections have costs to the host [17,18]. In the present study, we aimed to further examine the effects of persistent mono-infections with four distinct enteric RNA viruses—*Drosophila* A virus (DAV), *Drosophila* C virus (DCV), Bloomfield virus, and Nora virus—on a wide range of *D. melanogaster* physiological traits. Specifically, we studied survival rates, fertility (reproductive output, here defined as the number of viable offspring), bacteriome load and composition, locomotor abilities, and activity patterns. These biological traits are essential for understanding the host's overall fitness and its ecological interactions. Survival rates and reproductive output serve as key indicators of an organism's ability to thrive and contribute to the next generation [19]. The microbiome has been observed to influence various aspects of the host's physiology, such as nutrition status, immune function, behavior, and overall health [20–23]. Locomotor abilities and activity patterns are vital for behaviors such as foraging, mating, and avoiding predators, all of which are crucial for survival and reproduction [24]. By examining these traits, we highlighted the broader implications of persistent viral infections on insect biology.

## Results

### The dynamics of viral RNA levels over time are virus-specific

Our laboratory has previously established persistent viral mono-infections in *D. melanogaster* flies of the $w^{1118}$ genetic background using different viruses: DAV, DCV, Bloomfield virus, and Nora virus [25]. All these positive-sense single-strand

RNA viruses belong to different viral families and naturally infect *D. melanogaster.* They are transmitted primarily through orofecal contamination. In this work, we measured key biological traits over time in the infected and the uninfected control population (Fig 1A). To maintain continuous exposure to contaminated food while preventing excessive viral accumulation, flies from the different persistently infected lines were transferred to fresh food every other day. Persistent viral infections significantly reduce the survival of infected flies, as previously described (Fig 1B:). DAV infection has the most pronounced effects on survival, followed by DCV, Bloomfield virus, and Nora virus (Mantel–Cox test infected versus uninfected $p < 0.001$ for all the virus-infected populations). This is reflected by the time at which 50% of the population has perished, the median survival (Fig 1B): 25 days post-eclosion (dpe) for DAV, 28 dpe for DCV, 36 dpe for Bloomfield virus, and 52 dpe for Nora virus, compared to 63 dpe for uninfected flies. This information enabled us to establish different time points for further characterization of the infections: 1 dpe as an early time point, 12 dpe as an intermediate time point just before the onset of virus-induced mortality, and the median survival as a late time point specific for each condition. Additionally, we added an additional time point for DCV at 6 dpe, as mortality starts earlier for this virus. In the present study, these time points served as the basis for characterizing the impact of persistent viral infections on the physiology and behavior of infected hosts (Fig 1A).

To characterize the dynamics of viral load within the infected populations, we examined viral RNA levels in individual flies at the aforementioned early, intermediate, and late stages of infection. Viral loads were quantified by RT-qPCR by measuring viral RNA levels relative to the *Drosophila* housekeeping gene *Rp49*. Flies were classified as infected if their Ct values for viral RNA were below 35, with melting temperature profiles indicating the presence of a single amplicon. We observed distinct patterns of viral accumulation for each virus (Fig 1C, upper panels). DAV exhibited a rapid increase in viral RNA levels early post-eclosion, with a median increase of ~4 and ~5 $\log_{10}$ orders of magnitude at 12 and 25 dpe, respectively. In contrast, DCV displayed a relatively stable pattern in viral RNA levels over time, with no significant changes in viral loads. Interestingly, we noticed that DCV-infected individuals stably clustered into two distinct groups, displaying viral RNA levels several orders of magnitude apart. Bloomfield virus RNA levels rose steadily over the lifetime of infected individuals, increasing by 1 and 2 $\log_{10}$ orders of magnitude at 12 and 36 dpe, respectively. In contrast, Nora virus RNA levels remained consistent at the earlier time point but rose significantly by ~3 $\log_{10}$ orders of magnitude by the final time point (General Linear Model [GLM] letters over the conditions indicate grouping for multiple comparisons).

Our examination of viral RNA levels allowed us to estimate infection prevalence in the different populations over time (Fig 1C, lower panels). A viral infection is considered prevalent in a population if individuals continue to be infected over time. DAV and Nora virus were consistently detected in 100% of individuals, except for a slight reduction in Nora virus prevalence at the later stage (~91% of individuals infected at 52 dpe). Bloomfield virus was highly prevalent in the population at all time points, although not all individuals were infected at any given time, and infection prevalence varied from ~91% to ~72%. DCV was detected in all individuals at 1 dpe, and its prevalence in the population progressively decreased over time, with only ~36% of the individuals infected at the latest time point.

## Persistent viral infections impact viable offspring production

Persistent viral infections can reduce individual survival rates, which may consequently affect the viability of the population over time. To understand the broader impact of these infections, we investigated whether flies from populations with a prevalent viral infection produced altered numbers of viable offspring compared to flies from the uninfected population. For this experiment, we measured viable offspring production as the number of progeny that successfully emerged as adults. This metric is a crucial aspect of reproductive success and population dynamics. Any impact on the production of viable offspring would indicate that persistent infections influence additional critical biological traits related to fitness beyond the previously observed reduction in survival.

To explore potential changes in viable offspring production in the infected populations, 20 males and 20 females from each infected population or from the uninfected population were collected on the day of eclosion and placed together in

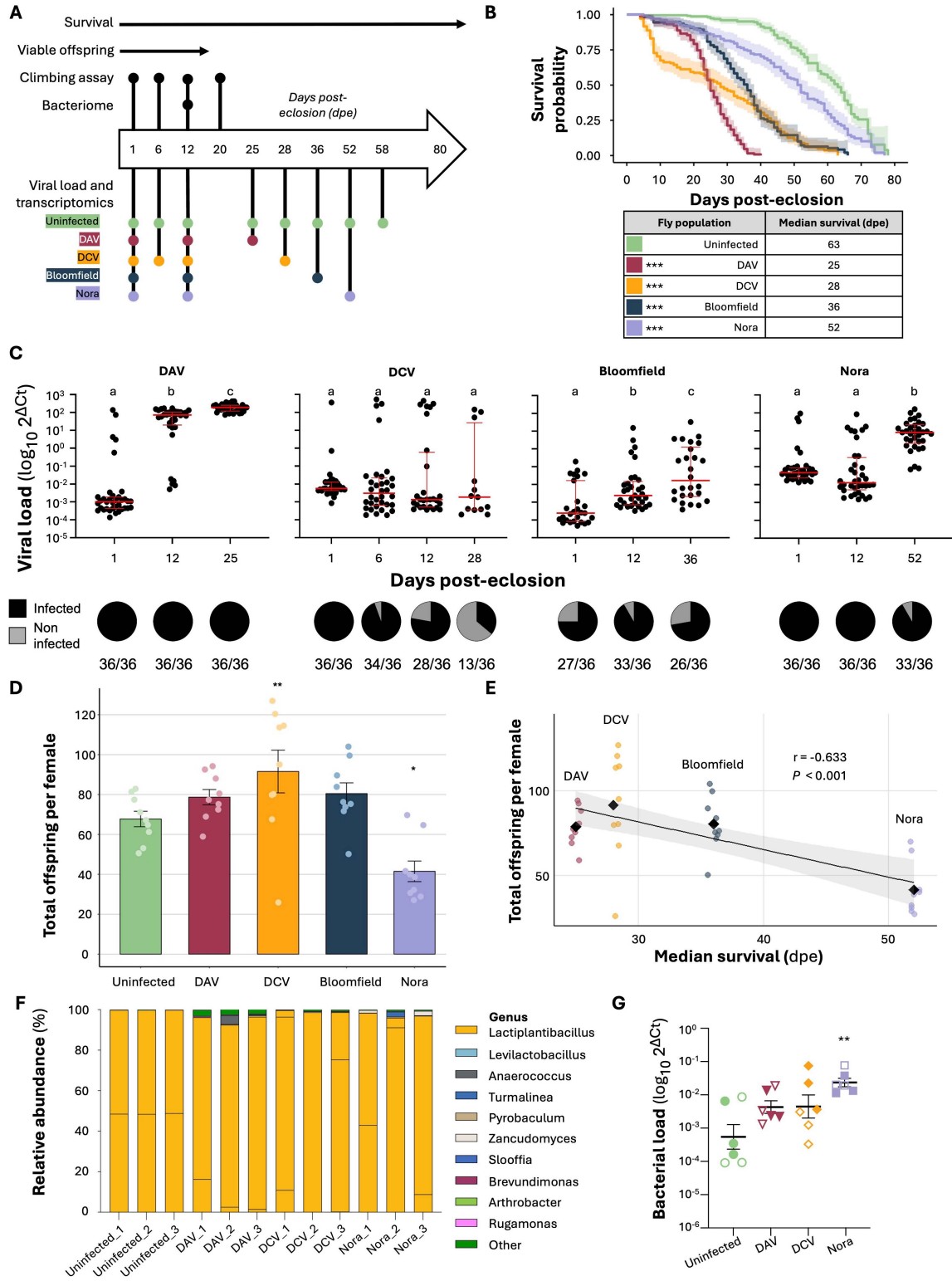

**Fig 1. Experimental set-up, survival, offspring production, viral dynamics, and bacteriome analysis of persistently infected fly populations.**
**A)** Experimental design. The large arrow indicates the days post-eclosion (dpe) during which specific experiments were conducted. The survival of persistently infected stocks has been monitored from eclosion until death across all stocks. The viability of offspring was assessed after a 24-hour mating

period, until no more embryos were laid (17 dpe). Black dots indicate the specific days when the climbing assay and the sampling for the bacteriome studies were performed for all stocks. At the bottom of the graph, regarding the transcriptomics experiment, each colored dot signifies the date of sample collection for RNA sequencing corresponding to each different fly stock. **B)** Left panel shows the survival of $w^{1118}$ flies uninfected (green) or persistently infected with DAV (red), DCV (yellow), Bloomfield virus (dark blue), or Nora virus (purple). Shaded regions indicate 95% confidence intervals. The data ($n = 180$ flies/condition in 3 independent experiments with 3 biological replicates/experiment) were obtained from Nigg and colleagues. Statistical significance was assessed via the Mantel–Cox test. Right panel indicates the dpe at which 50% of the population is deceased. **C)** Viral infection dynamics. The upper panel shows viral RNA levels for DAV, DCV, Bloomfield virus, and Nora virus in infected populations. Viral RNA levels were measured in whole individuals at various times post-eclosion. Only data from infected individuals is presented. The Y-axis denotes the relative viral RNA level as $\log_{10} 2^{\Delta Ct}$ between the viral RNA and the housekeeping gene, Rp49. The red bars represent the median values along with the interquartile range. For each virus, significance was calculated using a general linear model where the time point was the fixed factor; Tukey contrasts were used for post-hoc analyses. Letters over the bars indicate letter-based grouping for multiple comparisons. Bottom panel: viral prevalence showing the proportion of infected (black) or noninfected (gray) flies at the corresponding time point. Numbers below the pie charts represent the number of infected individuals over the number of individuals sampled. **D)** Cumulative viable offspring for each population, normalized over female number. Total viable offspring produced by each female was measured cumulatively across the different fly populations throughout the duration of the experiment, accounting for fly mortality. The Y-axis shows the total viable offspring per female and the X-axis the population studied. Significance was calculated using a general linear-mixed model where the virus status was the fixed factor and experiments a random effect; Tukey contrasts were used for post-hoc analyses. **E)** Relationship between the offspring and the survival of fly stocks. On the Y-axis the total offspring per female. On the X-axis, the median day of survival, in dpe, of each infected stock. Each dot on the graph represents a total offspring experiment. All infected samples from the same infected stock have the same median survival. For each virus-infected stock, the mean offspring is denoted as a black rhombus. Pearson's correlation coefficient $r = -0.633$ and $p$ value $< 0.001$ are depicted in the graph. **F)** Bacterial composition of pools of at least 20 females from the studied fly populations except for the one infected with Bloomfield virus. Females coming from the uninfected or persistently infected stocks were collected at 12 dpe, surface sterilized and their DNA was extracted. For each population, microbiome composition is expressed as the relative abundance in percentage of bacterial genera detected by sequencing of the full bacterial *16S* rRNA gene. Each color corresponds to a specific bacterial genus sequenced in the microbiome. Bacteria belonging to different operational taxonomy units (OTUs) but belonging to the same bacterial genus are displayed with the same color. For all the figure, \*\*\**p* < 0.001, \*\**p* < 0.01, \**p* < 0.05. **G)** Total bacterial load of pools of 10–20 females coming from the uninfected or persistently infected stocks except for the one infected with Bloomfield virus. Females coming from the uninfected or persistently infected stocks were collected at 12 dpe, surface sterilized and their DNA was extracted. The experiment was performed twice, with three pools per line analyzed in each experiment. Bacterial load was assessed via qPCR targeting the *16S* bacterial ribosomal gene. The Y-axis denotes the relative bacterial DNA level as $\log_{10} 2^{\Delta Ct}$ between the bacterial *16S* gene and the housekeeping gene, Rp49 (mean ± SEM). Shape fill indicates the replicate. Statistical significance was determined with a generalized linear model using $\log_{10} 2^{\Delta Ct}$ values and condition as fixed factors ($F_{(4,24)} = 26.63$, $p < 0.001$) and the replicate as a random factor. Tukey's adjustment was applied to post-hoc comparisons. Underlying data for all panels in this figure can be found in S1 Data.

fresh vials for 24 hours to allow mating. After 24 hours, males were discarded, and the 20 females were transferred into new vials with fresh food each day. Initial female density across all populations was uniformly established at 20 females per tube, as these conditions provide optimal surface area and total tube space, thereby minimizing crowding effects and promoting normal development. The viable offspring were counted daily as they emerged and the viable offspring count was normalized based on the number of live females remaining in each vial. This normalization allowed us to account for changes in the density of females per tube due to mortality over the course of the experiment.

Flies collected from all four virus-infected populations exhibited distinct patterns of offspring production compared to uninfected controls (S1 Fig). Uninfected flies exhibited a peak in viable offspring production at 1 dpe ($10.3 \pm 2.02$ adults emerging per female, mean ± SD), followed by a gradual decrease in offspring production until 6 dpe and a pronounced decline thereafter, reaching zero by 16 dpe. Similarly, populations infected with DAV and Bloomfield viruses also peaked in viable offspring production at 1 dpe ($11.5 \pm 2.07$ and $11.6 \pm 1.30$ offspring per female, respectively) and experienced a sharp decline in offspring production after 6 dpe. However, during early time points, DAV, and Bloomfield virus-infected populations produced more viable offspring per female than the uninfected control population (GLMM, DAV $P = 0.022$, Bloomfield $P = 0.031$). DCV-infected flies displayed a distinct pattern, producing significantly more viable offspring than the uninfected ones only at later time points (GLMM, $P < 0.001$). In contrast, Nora virus-infected flies consistently produced fewer viable offspring (GLMM, $P = 0.010$), with their offspring production peaking at 4 dpe ($5.28 \pm 2.15$ viable offspring per female).

Studying both daily and cumulative offspring production offers a comprehensive perspective on reproductive fitness. While daily offspring production captures temporal patterns and short-term reproductive dynamics, cumulative counts reveal the total reproductive output across a female's life span.

Analysis of cumulative offspring per female reveals that DCV-infected flies produced more viable offspring (91.5 ± 32.1) than the uninfected group (67.7 ± 11.6) (Fig 1D, GLMM, $P = 0.004$). In contrast, Nora virus-infected flies produced significantly fewer offspring per female (41.5 ± 15.5, GLMM, $P = 0.034$). Although DAV and Bloomfield virus-infected flies showed higher early production rates, their lifetime reproductive output did not differ significantly from uninfected controls. These results demonstrate that persistent viral infections have virus-specific impacts on reproductive fitness, with some viruses enhancing and others diminishing total offspring production.

Analysis of the relationship between total offspring production and female survival reveals a negative correlation across virus-infected stocks ($r = -0.633$, $P < 0.001$; Fig 1E). DAV and DCV infections resulted in higher offspring production but shorter lifespans, while Bloomfield and Nora virus infections showed the opposite pattern: lower reproductive output but longer survival. This trade-off suggests a potential reallocation of resources from reproduction to defense mechanisms in response to viral infections.

## Harboring persistent viral infections changes fly bacteriome

We next investigated whether persistent viral infections influence the fly bacteriome—the bacteria associated with the host. All fly populations in this study shared the same genetic background and were reared under identical conditions, except for the presence of different persistent viral infections. This controlled setup allowed us to attribute any observed differences in the associated bacteria to the presence of different viral infections.

Pools of at least 20 flies in three replicates were collected at 12 dpe, surface sterilized and their DNA was extracted. Microbiome composition was assessed by amplification and sequencing of the full *16S* rRNA bacterial gene through Oxford Nanopore sequencing. Despite good DNA yields for all samples, bacterial *16S* amplification was not successful for Bloomfield virus-infected flies, even when DNA was extracted from pools of 40 flies. We therefore could not determine the bacterial composition of Bloomfield virus-infected flies.

An average of 82,053 reads was obtained per sample (min: 25,673; max: 139,361). As previously described, we found a relatively low bacterial diversity in all the analyzed populations, with only 7–21 operational taxonomy units (OTUs) identified in our samples. α diversity analyses did not reveal any significant difference in the OTU diversity of uninfected or persistently infected flies (S3A Fig, pairwise Wilcoxon test on Chao1 and Shannon indexes, $P > 0.46$ for all comparisons). Similarly, beta diversity analysis did not identify specific differences in the bacterial community structure of the different fly lines (S3B Fig, PERMANOVA on infection, $F = 1.57$, $P = 0.24$). After taxonomical assignation of reads, we confirmed that the microbiome of all our *Drosophila* populations was dominated by *Lactiplantibacillus* bacteria characterized by a *16S* sequence identity higher than 99% (Fig 1F). The failed amplification of the *16S* gene of Bloomfield virus-infected flies suggested a significant difference in the total microbiome load of our fly populations. We thus determined the total bacterial load in whole flies by *16S* qPCR. Nora virus-infected flies showed a significant increase in total bacterial loads compared to uninfected flies. Similarly, flies persistently infected with DAV or DCV were characterized by a nonsignificant increase in their total bacterial loads (Fig 1G). Although we successfully amplify bacterial *16S* from Bloomfield virus-infected flies (with amplification cycles earlier than no-template controls), their Ct values closely matched those of extraction negative controls—blank samples processed alongside fly samples. This confirms that flies persistently infected with Bloomfield virus harbor a markedly reduced microbiota load.

Taken together, these data indicate that persistent viral infection does not influence the composition of the microbiome but significantly impacts the total bacterial load colonizing the fly.

## Persistent viral infections alter locomotor function in a virus-dependent manner

Locomotor function is a key indicator of an organism's overall health and physiological status. In flies, locomotion reflects both muscular and neurological capacity, making it a valuable marker for assessing the general impact of viral infections on host physiology.

We assessed locomotor function using the reverse geotaxis assay, which measures the flies' upward climbing instinct over a specified period [26]. The assay was performed on 20 mated females from the uninfected and the different infected stocks at 1, 6, 12, and 20 dpe. This final time point was chosen as a common late date for all fly stocks, occurring prior to the day of median survival for all of the infected populations. Flies were placed in an empty tube and collected at the bottom by gently tapping the tubes. After 3 s, a picture was taken and the position of each fly was recorded manually. Our observations indicated that persistent viral infections had a mixed effect on locomotor function over time (S2 Fig). DCV- and Nora virus-infected flies consistently showed impaired locomotion abilities compared to uninfected flies at almost all the tested time points (S2 Fig). In contrast, DAV- and Bloomfield virus-infected flies showed mixed locomotion patterns. DAV significantly reduced mobility at 1 dpe but significantly increased it at 6 and 20 dpe (GLMM, $P < 0.001$, $P < 0.001$, and $P = 0.011$, respectively). Bloomfield virus significantly reduced mobility at 1 dpe and increased it at 12 dpe (GLMM, $P < 0.001$ at both time points). When examining the fly distribution patterns along the tubes we obtained similar results, with a higher proportion of DCV- and Nora virus-infected flies found in the bottom of the tube at almost all time points (S2 Fig). DAV infection resulted in a higher proportion of flies climbing to the top of the tube at 6 dpe, but a higher proportion of flies at the bottom at 12 dpe (cumulative link mixed models [CLMM], $P < 0.001$ and $P = 0.001$). Bloomfield virus-infected flies were found more often on the bottom of the tube at 1 and 20 dpe, while they climbed more to the top at 12 dpe (CLMM, $P < 0.001$, $P = 0.003$, and $P = 0.032$, respectively). Taken together, these results indicate that persistent viral infections affect the climbing abilities of infected flies, with a general trend of lower climbing height, particularly in flies harboring Nora virus infections. These findings underscore the systemic impact of persistent viral infections on host physiology and behavior.

## Persistent viral infections alter the fly activity and sleep duration while preserving circadian rhythms

We continuously monitored fly movement over a 72-hour period under 12:12-hour light:dark conditions to assess the impact of persistent viral infections on activity patterns and sleep behavior. The movement data revealed that all experimental groups maintained clear circadian rhythmicity, with peaks in activity occurring at light-dark transitions and similar overall temporal patterns of activity (Fig 2A).

Analysis of sleep periods, defined as immobility bouts lasting at least 5 min ([27] Cirelli and Bushey, 2009), revealed significant differences between infected and uninfected flies (Fig 2B). During the night phase, flies infected with Bloomfield, DAV, or Nora virus spent significantly more time sleeping compared to uninfected controls (GLMM, all $P < 0.001$). DCV-infected flies showed no significant difference in night sleep compared to uninfected controls. Day-phase sleep patterns displayed a similar profile, with all infected flies exhibiting significantly increased sleep duration compared to uninfected controls (GLMM, all $P < 0.001$).

We further analyzed the intensity of movement during active periods (when flies were not asleep) to determine whether viral infections affected the quality of movement (Fig 2C). Bloomfield-infected flies showed significantly reduced movement intensity during their active periods compared to uninfected controls (GLMM, $P < 0.05$), suggesting that even when Bloomfield virus-infected flies were active, they moved less vigorously than uninfected flies. In contrast, flies infected with DAV, DCV, or Nora virus showed movement intensities during active periods that were comparable to uninfected controls, with no significant differences detected. Together, these results demonstrate that while persistent viral infections do not disrupt fundamental circadian rhythmicity in *Drosophila*, they significantly modify both sleep duration and movement intensity in a virus-specific manner.

## Host transcriptional response is virus-specific

To better understand the alterations induced by persistent viral infections, we examined the transcriptional response of flies at early (1 dpe), intermediate (12 dpe), and late stages of infection (at median survival times). For each time point and

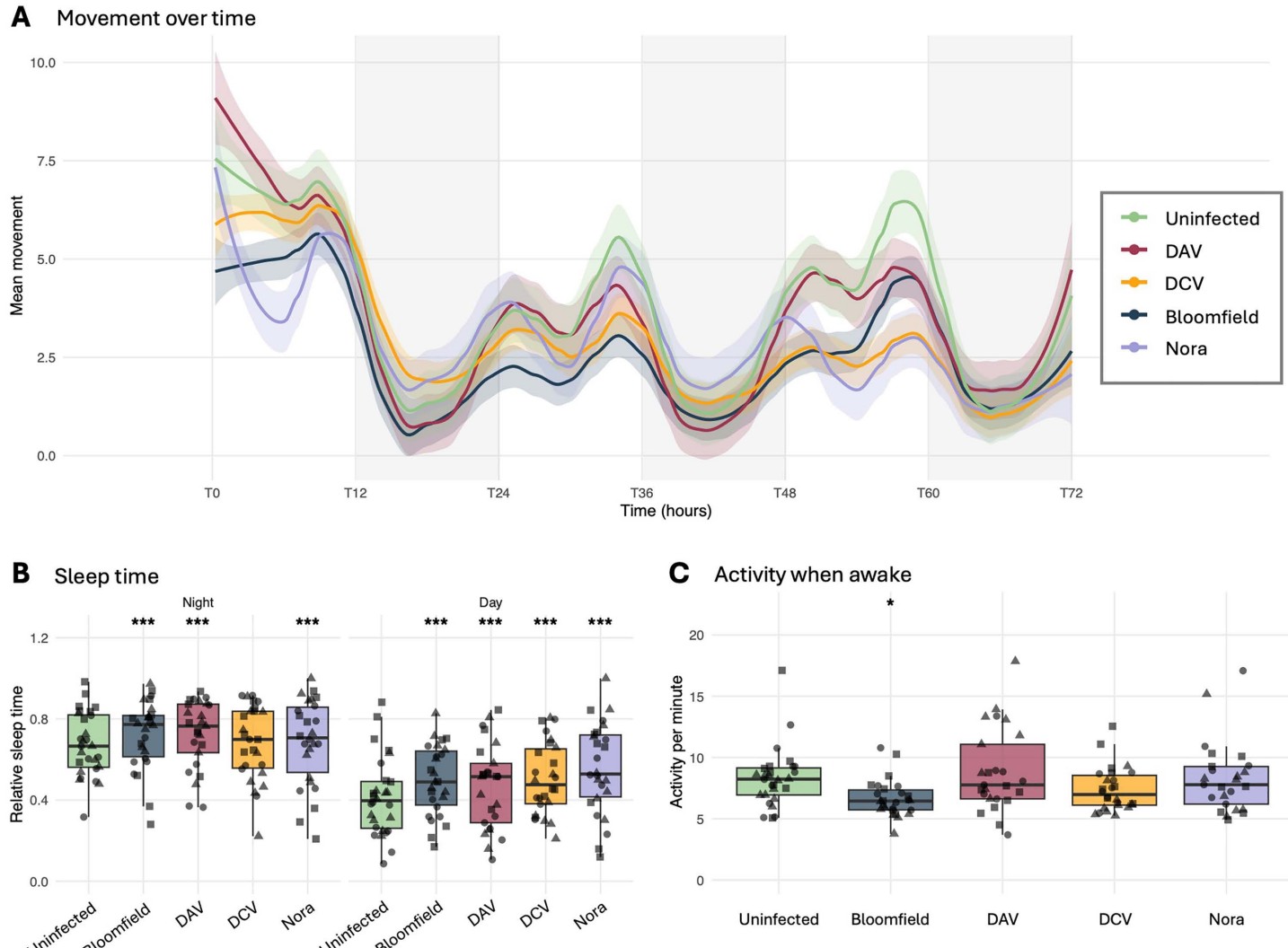

**Fig 2. Activity patterns and sleep behavior of persistently infected flies. A)** Mean locomotor activity over a 72-hour period under 12:12 light:dark conditions (T0–T72). Activity was quantified as infrared beam crossings per minute using *Drosophila* Activity Monitors. Colored lines represent different infection conditions (uninfected: green, Bloomfield virus: blue, DAV: red, DCV: yellow, and Nora virus: purple), with shaded areas indicating 95% confidence intervals. Gray shaded vertical bands indicate dark periods. Time is shown in hours, with T0 representing the beginning of the first light cycle after one night cycle of acclimation. **B)** Sleep time comparison during night and day phases. Sleep was defined as periods with no activity for at least 5 consecutive minutes. Relative sleep time is shown as proportion of total available time in each phase. Box plots indicate the median and interquartile range; each point represents an individual fly. Statistical significance compared to uninfected controls was determined using a linear mixed-effects model with experimental replicate as a random effect (***$P < 0.001$, **$P < 0.01$, *$P < 0.05$). **C)** Activity intensity during wake periods. Box plots show the average number of beam crosses per minute during nonsleep bouts. Each point represents an individual fly. Statistical significance compared to uninfected controls is indicated (*$P < 0.05$, ns = not significant) and was determined using a linear mixed-effects model with experimental replicate as a random effect. Data represent three independent experiments, each including 8–12 flies per condition. All experiments were conducted with 12-day-old mated female flies at 25°C under a 12:12-hour light:dark cycle. Underlying data for all panels and the code used to generated are deposited in Zenodo (https://zenodo.org/records/17175344).

condition, RNA was extracted from individual flies for individual measurements of viral load via RT-qPCR. Subsequently, aliquots of RNA from 3 to 5 infected individuals harboring similar viral load levels (Fig 1C) were pooled together for the generation of bulk RNA-seq libraries.

Principal component analysis revealed that each enteric virus elicited a distinct transcriptional response in the host (S4 Fig). Nora virus infection was characterized by the most unique transcriptional response at 1 dpe, with the differential regulation of several genes (S5 and S6 Figs). In contrast, at the same early time point, other viruses induced little changes in transcriptional profiles of infected flies and samples clustered together with uninfected flies (S4–S6 Figs). At a more advanced time point (12 dpe) the transcriptional profiles of DCV and Bloomfield virus-infected flies clustered together with uninfected flies, while Nora virus and DAV-infected flies were characterized by unique transcriptional responses (S4 Fig). By the days of median survival, all viruses had induced significant changes in transcriptional profiles (S4 Fig).

No common differentially expressed gene (DEG) was identified across all viral infections at any time point (S7 Fig). Furthermore, we could not identify a consistent set of host genes responding to the infection over time, thereby illustrating the differential impacts of viral infection on the host as the infection progresses (S7 Fig). Even at median survival, no common transcriptional profile was shared among all infections (S7 Fig). We also observed an increase in the number of DEGs as infection progressed for the flies persistently infected with DAV, DCV, and Bloomfield virus (S6 and S7 Figs). Many of the top DEGs identified at early and intermediate time points are uncharacterized (S6 Fig).

Focusing on the genes known to participate in antiviral immune responses, Nora virus elicited the highest levels of immune regulation at 1 dpe (Fig 3A). At 12 dpe, DAV had the highest upregulation of genes associated with immune defense and stress response pathways, indicating a potent antiviral response (Fig 3A). Interestingly, specific immune genes upregulated during DAV infection were downregulated during Nora virus infection, suggesting fundamentally different immune responses to these viruses (Fig 3A).

Analysis of genes involved in locomotion revealed significant differential expression in Nora virus-infected flies at 1 dpe. Notably, *Lsp2,* a gene reported to be involved in motor neuron axon and synaptic target inhibition [28], was upregulated in Nora virus-infected samples (Fig 3B). As infection progresses, we observed an increasing number of DEGs related to locomotion, with Nora virus infection showing an opposite transcriptional profile compared to DAV infection (Fig 3B). At 12 dpe, Nora virus infection upregulated genes involved in gravitaxis regulation (*CG3,857*), negative phototaxis (*Ptth*), light reception (*Rh5*), and neuronal function (*nompC, Pdf*) [29–33]. Furthermore, Nora virus infection induced the upregulation of *M6* (involved in several neurological processes, including positive phototaxis) and the downregulation of *fz* (neuronal function) and *Neto* (neuromuscular roles and hatching behavior) [34]. Conversely, DAV-infected samples showed upregulation of *Neto*, *esg* (related to stem cell maintenance and nervous system development), and *Mlc2* (flight musculature) [35–38]. DCV and Bloomfield virus infections induced more modest differential expression of locomotion-related genes. Taken together, these results indicate that the host response to persistent infection is virus-specific.

## Viral infection induced transcriptional changes even after viral clearance

While persistent viral infections remain prevalent at the population level, individual flies can clear the infection over time (Fig 1C; [25,39]). This was particularly evident for DCV, where all individuals were infected at 1 dpe, but more than half had cleared the virus by 28 dpe (Fig 1C). This provided an opportunity to examine whether viral clearance restores normal gene expression or leaves a lasting transcriptional signature.

Therefore, we analyzed the transcriptional response of flies that had cleared DCV infection by 28 dpe and found that it remained significantly altered compared to the never-infected control population (Fig 4A). Comparison of the transcriptional response of currently infected and cleared flies showed minimal differences between the two (Fig 4B), with most DEGs related to mitochondrial function. These included mitochondrial transfer RNAs and mitochondrial proteins such as *mt:ATPase8* (mitochondrial ATPase subunit), *l(3)neo43* (cytochrome c oxidase assembly), *Mcad* (mitochondrial fatty acid beta-oxidation), *mtSSB* (mitochondrial single-stranded DNA-binding protein), *ND-MWFE* (respiratory chain complex component), and *sea* (inner mitochondrial membrane carrier protein). There is a marked downregulation of mitochondrial genes in the cleared samples, which indicates that energy metabolism is upregulated during DCV infection.

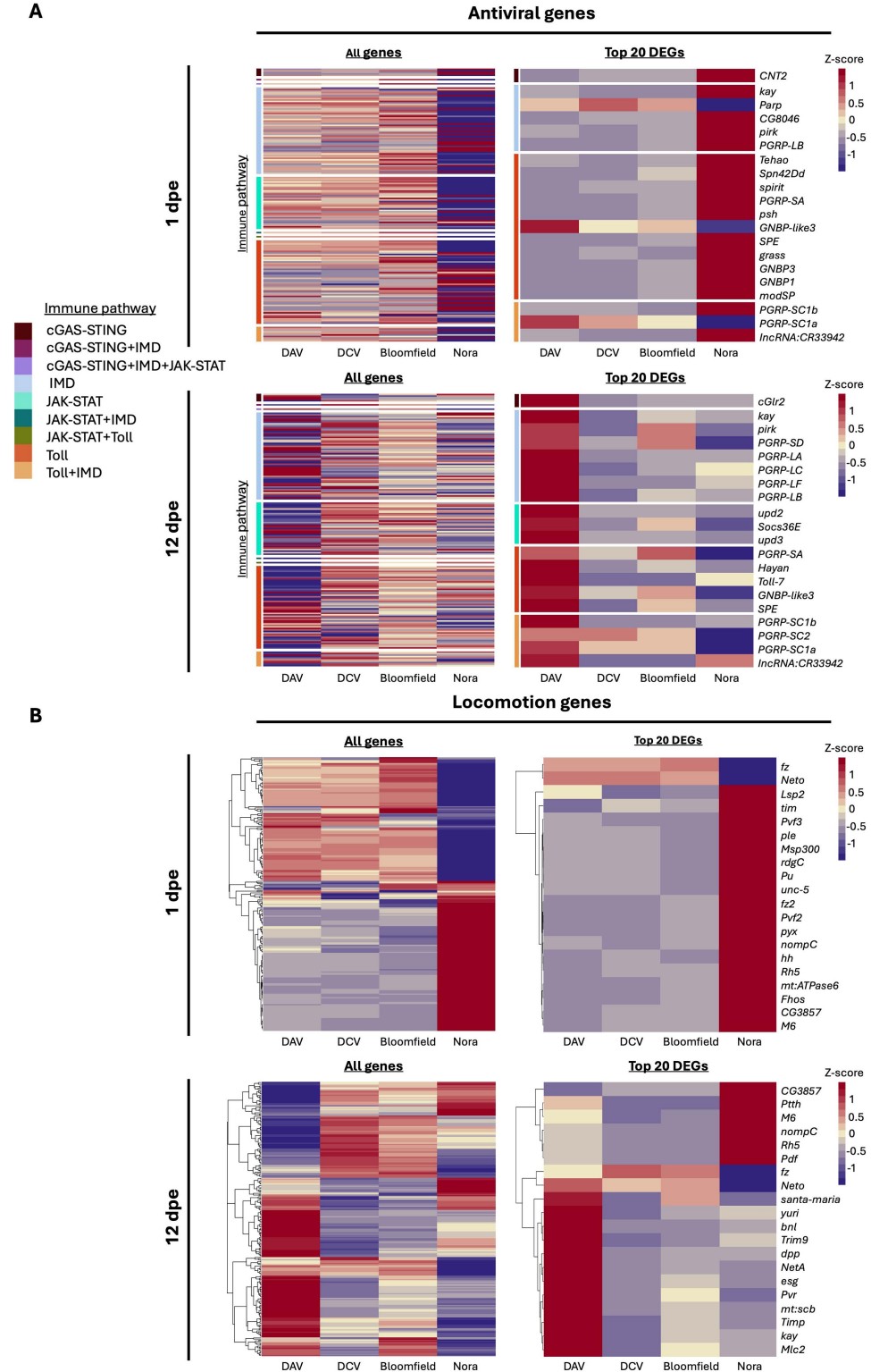

**Fig 3. Transcriptomic profiles of uninfected or persistently infected flies. A)** Heatmaps expressing the *Z*-scores of gene expression fold change of infected populations in comparison with uninfected ones. Genes are represented in the Y-axis and the infecting virus on the X-axis. For each virus, the results are calculated on the average of 4 samples. Top left: heatmap showing all the genes reported to be involved in antiviral immunity (subset

of 191 immune antiviral genes, S1 File) and their differential expression in our samples at 1 dpe. Top right: heatmap showing the top 20 differentially expressed immune genes at 1 dpe. Bottom left: heatmap showing all the genes reported to be involved in the insect's immune response and their differential expression in our samples at 12 dpe. Bottom right: heatmap showing the top 20 differentially expressed immune genes at 12 dpe. On the right side of each heatmap, a colored bar indicates the pathway in which the gene is reported to participate. **B)** Heatmaps showing all the genes reported to be involved in the insect's locomotion (a subset of 264 locomotion genes, S1 File) and their differential expression in our samples at 1 dpe1. Top left: heatmap showing the top 20 differentially expressed locomotion genes at 1 dpe. Top right: heatmap showing all the genes reported to be involved in the insect's locomotion and their differential expression in our samples at 12 dpe. Bottom left: heatmap showing the top 20 differentially expressed immune genes at 12 dpe. Bottom right: heatmap showing the top 20 DEGs at 12 dpe. Underlying data for all panels in this figure can be found in S2 Data.

Both infected and virus-cleared flies maintained significant transcriptional differences compared to never-infected controls, including sustained upregulation of immune response genes (Fig 4D). These results indicate that viral infections induce long-lasting or potentially permanent alterations in host gene expression, particularly affecting mitochondrial and immune functions, that persist even after viral clearance.

## Discussion

Our study provides a comprehensive examination of how persistent viral infection impacts key biological traits in *D. melanogaster*. By studying mono-infections with four different enteric RNA viruses, we reveal a complex interplay between viruses and their hosts that affects survival, reproductive output, bacteriome load, locomotor abilities, and transcriptional profiles.

We observed a trade-off between survival and reproductive output. This pattern suggests a compensatory mechanism in which, when faced with specific stressors, such as viral infections, the host organism may prioritize reproduction over immunity [40,41]. Indeed, for DCV, DAV, and Bloomfield virus, we noted reduced survival but increased offspring production in infected flies, indicating a redirection of resources from surviving the infection to reproduction. Conversely, for Nora virus infections, resources may be redirected primarily toward surviving the infection, as flies infected by this virus exhibit the smallest impact on survival but a significant reduction in offspring production. These diverse interactions correspond with the unique host transcriptional changes induced by each virus. Notably, Nora virus causes a more substantial disruption to the transcriptome, suggesting a higher immune investment in combating the infection. This reduced survival appears to result from an acceleration of biological aging, with our transcriptional dataset revealing that more pathogenic viruses induce more rapid biological aging [42]. The differing impacts on offspring production observed across various viral infections align with previous findings [18]. For instance, it has been reported that DCV-infected flies show increased fecundity compared to uninfected mothers [43]. Of note, our protocol for measurement of offspring production involved transferring parental females to fresh food daily. Whether the frequency of transfer to fresh food has an impact on the dynamics of viral infection in the parental females is currently unknown and therefore this caveat should be taken into account when considering our data regarding offspring production.

The fitness costs we observed have important evolutionary implications for virus-host dynamics. Persistent viral infections that reduce survival while altering reproductive output could drive strong selection for resistance or tolerance mechanisms in *Drosophila* populations [44,45]. Indeed, natural *Drosophila* populations show considerable genetic variation in susceptibility to viral infections, suggesting ongoing host-pathogen coevolution [46–49]. The virus-specific effects we documented may explain why different antiviral mechanisms have evolved in insects, as each virus likely imposes distinct selective pressures on host populations. Moreover, the persistence of transcriptional changes even after viral clearance suggests that initial exposure events could have lasting effects on host fitness. These findings highlight that persistent infections may shape host evolution differently than acute infections, warranting further investigation into the consequences of persistent viral infections on population genetics in natural *Drosophila* communities.

The viruses studied in this work are enteric viruses, meaning that they are naturally transmitted via the fecal-oral route, infecting the organism through the gut. Our observations show that persistent enteric infections do not affect microbiome

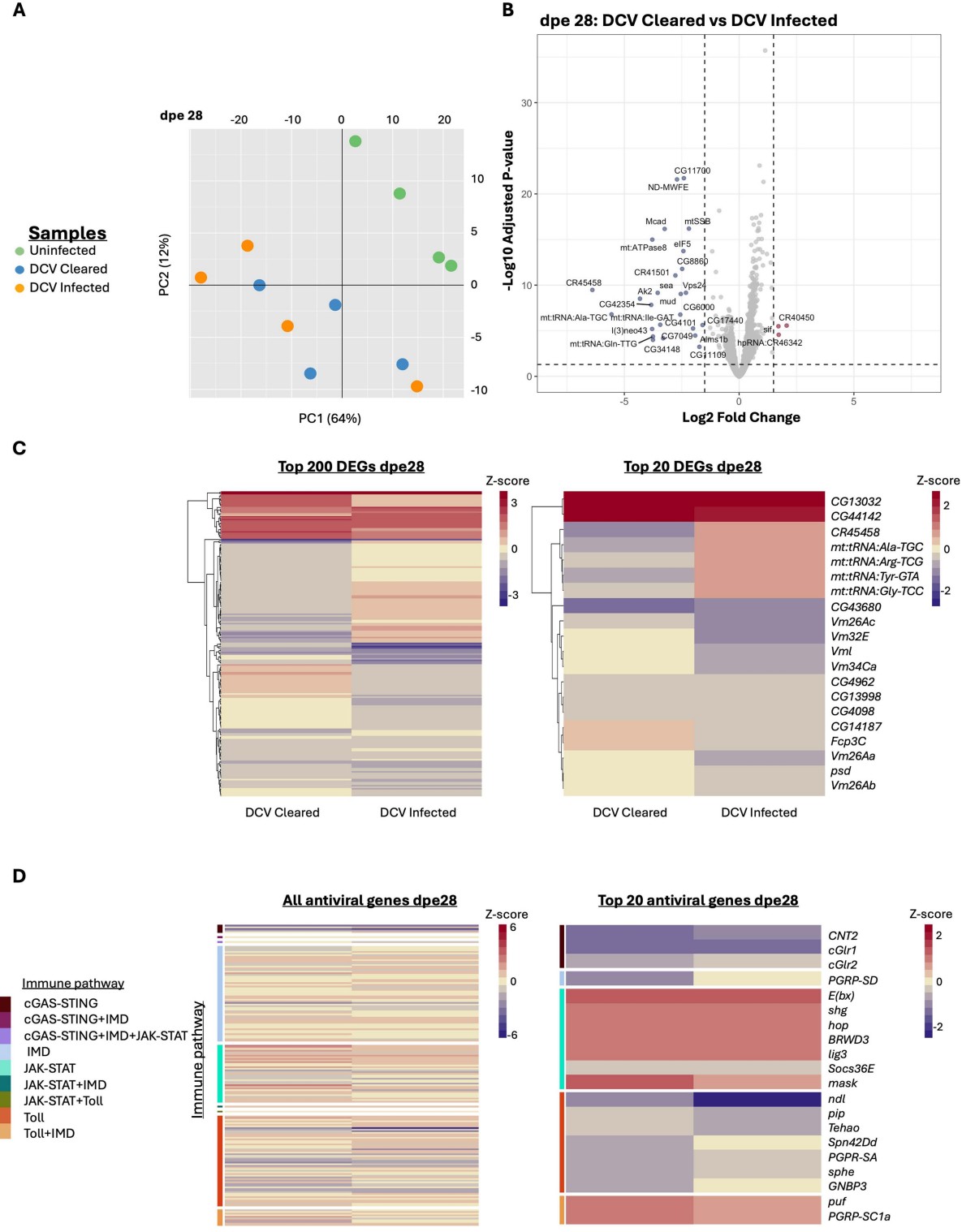

**Fig 4. Transcriptomic profile of flies at 28 dpe that were infected with DCV but cleared the viral infection. A)** Principal component analysis of uninfected samples, DCV Infected, and DCV Cleared. On the X-axis, we indicate the first principal component (PC1) and the percentage of variance explained by this component in brackets. On the Y-axis, we indicate the second principal component (PC2) and the percentage of variance explained

by this component in brackets. **B)** Volcano plot indicating the differentially expressed genes (DEGs) between DCV Cleared samples and DCV Infected ones. Genes significantly up- (in red) or down- (in blue) regulated. In gray, we represent those genes with differential expression values lower than 1.5 $\log_2$ fold change and *p*-adjusted values lower than 0.05. These thresholds are indicated by dotted lines in the graphs. The names of the DEGs are shown. **C)** Heatmaps representing the differential expression patterns between DCV Infected and DCV Cleared samples when compared to the Uninfected of the same age. Heatmap on the right shows the top 200 DEGs in our samples in terms of absolute $\log_2$ fold change value at dpe 28. Heatmap on the left shows the top 20 DEGs at dpe 28; gene names are indicated on the left. **D)** Heatmaps representing the differential expression patterns for reported immune genes between DCV-infected and DCV Cleared samples when compared to the Uninfected of the same age. Heatmap on the left shows all the genes reported to be involved in the insect's immune response and their differential expression in our samples at dpe 28. Heatmap on the right shows the top 20 differentially expressed immune genes at dpe 28; gene names are indicated on the left. On the right side of each heatmap, a colored bar indicates the pathway in which the gene is reported to involved. Underlying data for all panels in this figure can be found in S3 Data.

composition but significantly alter the size of the bacterial community present within the flies, which is mainly located in the gut. However, the bacteriome of laboratory-reared flies does not fully recapitulate that of wild flies [50]. The microbiome can modulate immune responses, nutrient absorption, and overall host physiology and thereby affect the outcome of viral infections [23,51,52]. Hence, some of the effects observed in gene expression or locomotor activity could be partially explained by the microbiome changes induced by the viral infections. However, it has been observed that for infections with DAV and Nora virus [53], the relative impact on fly survival is largely independent of the microbiome.

The enteric viruses studied here primarily infect the gut. During DAV infection, the guts of *D. melanogaster* exhibited overproliferation of intestinal stem cells, dysplasia, and a loss of gut integrity. However, our observations indicate that their impacts extend beyond the gut, as seen in the locomotion results. The effects on locomotion reflect changes in metabolic and/or neurological status [54] and may have profound implications for fly behavior and ecology. Impaired locomotion can affect activities such as foraging, mate searching, and dispersal, consequently influencing the fly's ability to colonize new locations and disseminate associated microbes. This has important implications for disease vectoring and the spread of pathogens. The systemic impact of enteric infections in flies has also been observed for enteric bacteria [55]. One potential explanation for this is the virus spreading from its primary location in the gut to other tissues. Such spreading has been documented for DCV [56] and DAV. Further studies are required to understand the extent and mechanisms of this viral dissemination. Another potential explanation for the systemic effects of enteric infections is the dissemination of immune signaling molecules. Molecules released in response to localized gut infections can have far-reaching effects on distant organs and tissues. For instance, studies have shown that gut-derived inflammatory signals can influence brain function and behavior, a phenomenon known as the gut-brain axis [57,58]. This interorgan communication can modulate various physiological processes, including metabolism, immunity, and even cognitive function. Therefore, the systemic impacts of enteric viruses may be mediated not only by direct viral spread but also by the immune signaling molecules that circulate throughout the host.

Our analysis of activity patterns and sleep behavior further demonstrates the systemic effects of persistent viral infections. While all viral infections preserved the flies' fundamental circadian rhythmicity, they significantly altered sleep duration, with most infected flies spending more time sleeping. This suggests that persistent infections affect specific sleep regulatory pathways rather than disrupting the core circadian clock mechanism [59]. The increased sleep duration may represent a host-driven response similar to the "sickness behavior" observed in vertebrates [60,61], potentially serving as an energy conservation strategy during infection. Interestingly, only Bloomfield virus-infected flies exhibited reduced movement intensity during active periods, suggesting virus-specific effects on neuromuscular and/or metabolic function. These findings align with previous studies showing that immune challenges alter sleep patterns in *Drosophila* [62–64] and highlight how different viral pathogens may interact with distinct host pathways. The virus-specific alterations in sleep and activity patterns likely reflect differences in how each virus affects energy metabolism, neural signaling, or immune activation, further emphasizing the complex and multifaceted nature of host-pathogen interactions. The lasting transcriptomic alterations we observed in DCV-cleared flies provide molecular evidence for these persistent physiological changes. The downregulation of mitochondrial genes in cleared flies compared to currently infected flies suggests that mitochondrial

reprogramming occurs during active infection and persists after viral clearance. This shift could serve dual purposes: meeting increased energetic demands during infection while simultaneously restricting cellular resources available for viral replication. The importance of mitochondrial function in infection outcomes is further supported by recent evidence that mitochondrial DNA variants can confer broad protection against diverse pathogens, with protective mitotypes showing upregulated mitochondrial respiration genes even in uninfected states [65].

Our study has certain limitations. Although infections were established at eclosion, not all flies remained infected throughout their life span. While DAV, Nora virus, and Bloomfield virus maintained high prevalence (~100%, 92%, and 72% of individuals, respectively), most individuals in the DCV-infected population cleared the virus by the time of median survival. Nevertheless, flies that cleared DCV showed lasting transcriptional changes, suggesting long-term effects reminiscent of post-acute infection syndromes [66]. Additionally, viral loads varied among individuals within the same infected stock, driving variation in life history traits within populations. However, viral load quantification requires tissue homogenization and fly sacrifice, preventing longitudinal studies of the same individuals across behavioral and fitness assays. Regarding measurements of offspring production, daily transfers could theoretically reduce viral accumulation in the environment compared to a 2-day interval (as was employed for the survival curve and viral RNA accumulation experiments), potentially influencing infection dynamics. However, our 1-day protocol was selected to maximize precision in offspring quantification with fine temporal resolution while avoiding larval overcrowding and maintaining natural viral load fluctuations. Finally, our study focused on females of a single genetic background, whereas both sex and genetic background are known to influence the effects of viral infections in *Drosophila* [48,67].

Our work presents a comprehensive multi-parameter assessment of how persistent viral infections affect *Drosophila* biology, providing valuable reference datasets and analytical frameworks for the research community. The primary contributions of our study include: (1) detailed transcriptional profiles across different timepoints and infection states that serve as reference resources for future investigations, (2) integrative analyses connecting molecular signatures to physiological and behavioral phenotypes, and (3) demonstrating virus-specific impacts on gut microbiome. Our study offers the first holistic view of how different persistent infections affect interconnected biological processes from gene expression to whole-organism fitness. The comparative datasets we provide enable researchers to examine the relationships between transcriptional responses, microbiome changes, and physiological outcomes across different viral infections. These reference resources support future investigations into viral persistence mechanisms, host-pathogen coevolution, and the broader ecological implications of persistent infections in insect populations. In conclusion, our study demonstrates that persistent viral infections significantly impact key biological traits in *D. melanogaster*. Our work adds to the growing evidence that persistent infections profoundly shape host fitness and physiology through complex and diverse mechanisms.

## Materials and methods

### Fly stocks

For all experiments, we used *Wolbachia*-free *w^1118* flies maintained on a standard cornmeal diet. This diet was prepared by combining 440 g inactive dry yeast, 440 g corn meal, and 60 g agar in 6 L of osmotic water. The mixture was autoclaved and after cooling, 150 ml moldex solution (20% methylhydroxybenzoate) and 29 ml propionic acid were added as preservatives. All experiments were done at 25°C under a 12:12 light:dark cycle. All fly lines were cleaned of possible persistent infections (viruses and *Wolbachia*) as described previously (Merkling and van Rij, 2015).

### Viral infections

Persistently infected *w^1118* flies were established following the method described by Nigg and colleagues [25] and are the same stocks used in the work of Nigg and colleagues. Persistent infections of four RNA viruses were established in *w^1118* *D. melanogaster* using tailored methodologies. For DCV, a persistently infected bw1; st1 Ago3t3/TM6B; Tb+ line (Bloomington #28270) was used to environmentally transfer DCV to naïve *w^1118* flies. Infected donors were housed in a fresh

vial for 3 days; after their removal, *w1118* flies were exposed to the contaminated environment for 3 days. The F0 generation was transferred to a new vial, and their F1 progeny were confirmed as persistently infected. For DAV, the Australian isolate (DAV$_{HD}$ from the van Rij lab) was injected into *w1118* adults (50 nl/fly). Identical environmental transfer followed: injected flies contaminated a vial for 3 days before naïve *w1118* exposure. Persistently infected F1 progeny were isolated and maintained. Bloomfield virus was identified in a contaminated Dipt-GFP stock (BDSC Cat#55709). Homogenate from these flies was filtered (0.22 µm) and injected into *w1118* adults. Injected F0 flies were removed after 3 days; F1 and F2 progeny underwent sequential generational passages (5–9 days/vial) to establish persistence in F3 adults confirmed via RNA sequencing. The Nora virus-persistent line originated from a *w1118* line persistently infected with Nora virus that was gifted by Dr. Stefan Ameres. Upon receipt, we repeatedly backcrossed the infected *w1118* flies to the *w1118* line maintained in our laboratory. Subsequently, we confirmed that the backcrossed flies retained persistent infection with Nora virus by RNA sequencing. The stocks were maintained at 25°C under a 12:12 light:dark cycle.

The presence or absence of persistent infections was determined by RT-PCR with specific primers for Nora virus, DAV, and DCV (S2 Table). In addition, total RNA from persistently infected adults was extracted and sequenced. The resulting reads were mapped to a database of all known *Drosophila* viruses (https://obbard.bio.ed.ac.uk/data.html) confirming that only the desired virus was present [25].

### Survival analysis

Survival analyses of the flies coming from persistently infected stocks were conducted using 20 female flies/replicate sorted at 1 dpe. Male and female flies were collected on the day of eclosion and left to mate for 24 hours. Subsequently, only female flies were sorted in new tubes containing 20 females each. Each tube was considered a biological replicate, and a total of nine biological replicates (from three independent experiments) were analyzed. Survival was monitored daily by counting the number of dead flies in each vial, and flies were transferred to new vials every 2 days. Fly survival was assessed from the day of adult eclosion until death.

### Viral quantification

Individual female flies were collected in 300 µl of TRIzol reagent (Invitrogen, 15596026). The samples were homogenized using a pestle, and RNA was extracted following manufacturer's instructions, supplemented with GlycoBlue (Invitrogen, AM9516) during the precipitation process. RNA concentrations were measured using the Qubit RNA BR Assay Kit (Invitrogen, Q10211). Following extraction, 4 µl of each RNA sample were treated with DNAse I (Roche, 04716728001), and 2 µl of the final volume were used in the RT reaction. RT was performed using random primers and Maxima H Minus Reverse Transcriptase (Thermo scientific, EP0751) following manufacturer's instructions. The resulting cDNA was diluted 1:10 with MilliQ water. The qPCR was performed using Luminaris Color HiGreen qPCR Master Mix (Thermo Scientific, K0374) in a QuantStudio 7 flex machine (Thermo Scientific). The reactions were done in triplicates of 10 µl each and using the corresponding set of primers targeting a replication protein for each virus (S1 Table). The cycling conditions were as follows: 2 min at 50°C, 10 min at 95°C, followed by 40 cycles of 15 s at 95°C and 60 s at 60°C. A standard melt curve analysis was performed after the cycling protocol. For the analysis of the results, the threshold for viral RNA detection was set to 35 Ct (i.e., if a sample had Ct higher than 35 it was considered noninfected). The Ct values of the virus for a sample were normalized against the Ct value of the housekeeping gene *Rp49* for that sample ($\Delta$Ct). The $2^{-\Delta Ct}$ values were calculated per sample and their $\log_{10}$ plotted.

### Viable offspring production

To assess viable offspring production, 20 female and 20 male flies from infected or uninfected stocks were left to mate in clean fly vials for 24 hours after eclosion. After this period, males were discarded, and the mated females were transferred to new tubes daily until no more progeny was produced (17 days). All tubes were kept at 25°C under a 12:12 light:dark

cycle, and the number of adult flies (i.e., the viable progeny) in each tube was counted after 14 days. The number of offspring was normalized to the number of adult females placed in each tube daily to account for those that did not survive throughout the test duration.

## Climbing assays

Climbing assays were performed following the method described by Barone and Bohmann [68]. Briefly, a 9 cm empty tube was used, and 20 female flies were placed inside. With the cotton cap on, tubes had a maximum climbable height of 7.2 cm. The tubes were tapped three times over the span of 3 s, and the climbing height was assessed by taking a picture 3 s after the last tap. Flies' height was assessed manually on the pictures using a metric guide on the back of the tubes. For categorical classification, flies were considered "Top" if they surpassed 6.2 cm, "Bottom" if they were between 0.0 and 1.0 cm, and "Middle" if they were between 1.0 and 6.2 cm.

## Behavioral analysis

We conducted behavioral analyses using DAM5H *Drosophila* Activity Monitors (Trikinetics, Princeton, MA, USA). Using the DAM5H we tracked the locomotor activity of individual flies housed in a 5 mm-diameter, 80 mm-length tube. The system records activity when a fly crosses one of 15 infrared beams of each tube. The monitor quantifies movement by registering infrared beam disruptions as activity "counts." A fly that moves continuously through a beam zone or returns to the same beam multiple times generates several counts, whereas a fly that simply enters a beam area and stops moving is recorded as only a single count. The count data were registered per minute and were used to evaluate the locomotion activity and to identify sleep periods, which were defined as immobility bouts lasting at least 5 min. All behavioral experiments utilized 12-day-old mated female flies. Flies were individually housed in DAM tubes containing standard Bloomington cornmeal diet. All experiments were conducted at 25°C under a 12:12 light:dark cycle. To allow for acclimation, flies were placed in the monitoring tubes 1 hour before the start of a dark cycle. Data collection began at the onset of the following light cycle (designated as T0) and continued for three complete cycles (until T2). We performed three independent experiments, each including 8–12 flies per experimental condition.

## Microbiome composition

Female flies from uninfected or persistently infected stocks were collected at 12 dpe. External microbes were removed by surface sterilization using subsequent washes in 10% bleach for 5 min, 70% ethanol for 5 min and three rinsing steps in sterile water. Since the fly microbiota is fairly stable in populations reared together, microbiota composition was analyzed in three pools of 20 flies per condition. Flies were placed in a lysis buffer (10 mM Tris-HCl (pH 8), 26 mM EDTA, 0.5% SDS, and 5 mg/mL lysozyme) and homogenized with sterile pestles. Blank samples consisting of lysis buffer were processed alongside the flies and served as extraction negative controls. After incubating the lysates for 1 hour at 37°C, total DNA was extracted using the DNeasy Blood and Tissue Kit (Qiagen) following the manufacturer's instruction. For each sample, 5 µL of undiluted genomic DNA was used as template in a 20 µL final reaction volume for amplification of the *16S* rRNA gene. The primers containing locus-specific sequence 8/27_Forward and 1492_Reverse were employed (S1 Table). PCR amplification was carried out for a total of 35 cycles. No visible amplicons were obtained from extraction negative controls. The resulting amplicons were used for library preparation with the Ligation Sequencing Kit V14 (SQK-LSK114, Oxford Nanopore Technologies) following the manufacturer's instructions. Subsequently, the ONT DNA libraries were sequenced on a PromethION 2 Integrated apparatus using a FLO-PRO114M flowcell. The raw ONT sequencing data were base-called (super-accurate algorithm), demultiplexed and sequencing adaptor trimmed using the software dorado (version 7.3.11). The produced long-reads in fastq format were quality checked using the software NanoPlot (version 1.42.0) and then quality filtered using the software filtlong (version 0.2.1), discarding any long-reads with an average quality score below Q17 and read lengths not in the range of 1,200–2,300 bp. The surviving long-reads were in-silico PCR

amplified to verify the locus using the usearch software (version v11.0.667), then clustered using the software isONclust (version 0.0.6.1). For each cluster, a consensus sequence was generated to represent it as an OTU. OTU sequences were compared to the reference sequences of the NCBI targeted loci DB (https://www.ncbi.nlm.nih.gov/refseq/targetedloci/ - downloaded 15 Feb. 2024). Taxa were predicted and confidences were calculated using the SINTAX algorithm implemented in the usearch software (version v11.0.667). DNA library construction, sequencing, read processing, and taxonomy identification described in these sections were performed by Microsynth AG (Balgach, Switzerland). Alpha and beta diversity measures were estimated using the Chao1 and Shannon indices (alpha diversity) and the

Bray-Curtis dissimilarity (beta diversity) in R using the phyloseq package (version 1.42.0). Both alpha and beta diversities were calculated on microbiota composition at the OTU level. Pairwise Wilcoxon tests were used to determine differences in alpha diversity measures among the different fly lines, while PERMANOVA was used to determine the effect of the infection status on Bray–Curtis dissimilarities. The barplot was produced in Prism (version 10.4.2). As *Lactiplantibacillus* OTU sequences were characterized by a sequence identity >99%, we decided to display the bacterial genus without indicating the bacterial species. Raw sequencing reads will be made available upon publication.

## Bacteriome load

Female flies from uninfected or persistently infected stocks were collected at 12 dpe. External microbes were removed as described above. Ten to twenty flies per condition were collected in two independent experiments. Total DNA was extracted as described above. Bacterial DNA was quantified via amplification of the *16S* rRNA bacterial gene and the fly ribosomal gene *Rp49* via qPCR using the Luminaris Color HiGreen qPCR Master Mix (Thermo Scientific, K0374) in a QuantStudio 7 flex machine (Thermo Scientific). Primer sequences are listed in S1 Table. Ct values were retained only if they were lower than those of both no-template and extraction blank controls. Total bacterial load was calculated via the ΔCt method.

## Transcriptome profiling by high-throughput sequencing

Raw sequencing reads can be found in the Sequence Read Archive under the BioProject PRJNA1235228. For each condition, four samples were sequenced. Each sample was made by pooling RNA from five individuals (3 ng of total RNA per individual), except for DCV at 28 dpe, where the four pools were made from three individuals each. RNA-seq libraries were prepared from 15 ng of pooled RNA using an NEBNext Ultra II RNA Library Prep Kit for Illumina (New England Biolabs, E7770L) with NEBNext Multiplex Oligos for Illumina (Dual Index Primers Set 1) (New England Biolabs, E7600S). All sequencing was performed on an Illumina NextSeq 500 instrument using a NextSeq 500/550 High Output Kit v2.5 (75 cycles) (Illumina, 20024906). Raw sequencing reads were mapped to the *D. melanogaster* genome release dmel-r6.56 with STAR version 2.7.11b [69]. Feature counting was performed with HTSeq version 0.11.2. [70], and differential gene expression analysis was performed with DESeq2 in R Studio (version 2023.03.1). Differential gene expression analysis results can be found in S2 Data. Log$_2$ fold changes were transformed into *Z*-scores using the "ashr" package, and only adjusted *p*-values were considered for analysis.

## Statistical analysis

All statistical analyses were conducted using R version 3.6.1 within the Rstudio development environment version 2024.04.2 + 764. The level of significance was set at $P < 0.05$ and symbols indicate the significance of differences between conditions: \***$P < 0.001$; \*\*$P < 0.01$; \*$0.01 < P < 0.05$.

To account for the block effect, we employed linear mixed-effects or general linear mixed-effects models. Generalized linear models (GLM) were fitted to the data using the g*lm f*unction from the "stats" package. Generalized linear mixed models (GLMM) were fitted to the data using the *lmer f*unction from the "lme4" package version 1.1-35.5. CLMM using the *clmm* function from the "ordinal" package version 2023.12-4.1. *Post-hoc* pairwise comparisons were performed using the

*emmeans* function from the "emmeans" package version 1.10.5. Tukey's or FDR correction was used to adjust for multiple comparisons. The letter-based grouping for pair-wise comparisons was created using the *cld* function from the "multcomp" package version 1.4-26. The specific statistical methods applied to each figure are detailed in the legend of the corresponding figure.

## Supporting information

**S1 File. Genes associated with antiviral immune or locomotion function.**
(XLSX)

**S1 Data. Underlying data for Fig 1 and S1 and S3 Figs.**
(XLSX)

**S2 Data. Underlying data for Fig 3 and S4, S5, S6, and S7 Figs.**
(XLSX)

**S3 Data. Underlying data for Fig 4.**
(XLSX)

**S4 Data. Underlying data for S2 Fig.**
(XLSX)

**S1 Table. Primers used in qPCR step of RT-qPCR for viral load quantification.**
(DOCX)

**S2 Table. Primers used for RT-PCR to assess the presence of viruses and *Wolbachia*.**
(DOCX)

**S1 Fig. Impact of persistent viral infection on offspring production.** Daily viable offspring per female. Number of viable offspring produced by female flies uninfected (green) or persistently infected with DAV (red), DCV (yellow), Bloomfield virus (dark blue) or Nora virus (purple) after one day of mating (20 females paired with 20 males). After one day, the males were discarded, and the females were transferred to new vials with food each day. Mated flies were kept for subsequent days, and the viable offspring were counted. The number of viable offspring was normalized by the number of alive females eggs in each tube. Day 0 represents the day of eclosion and mating. Experiments were conducted on 20 females paired with 20 males in 9 replicates. Significance was calculated using a general linear-mixed model where the time point was the fixed factor and experiments a random effect; Tukey contrasts were used for post-hoc analyses. Significance values are *** $P < 0.001$, ** $P < 0.01$, * $P < 0.05$.
(TIF)

**S2 Fig. Impact of persistent viral infection on locomotion.** Climbing assay results of uninfected flies (green) or flies persistently infected with DAV (red), DCV (yellow), Bloomfield virus (dark blue) or Nora virus (purple). **A)** Quantitative height climbed by individual flies. Climbing assay results for days 1, 6, 12, and 20 post-eclosion. Results of 9 different biological replicates with 20 female flies. At each time point and replicate, six pictures were taken, and the height of each fly in the tube was recorded. Each symbol on the graph represents the recorded height of a fly for each picture. Boxplots are shown on the left side of each stock, indicating the median and interquartile range of the heights climbed by the flies. The dotted line represents the median height climbed by the uninfected population. The half-violin of each plot shows the density of the values. Significance was calculated using a general linear-mixed model where the virus status and the time point were the fixed factors and experiments a random effect; Tukey contrasts were used for post-hoc analyses. **B)** Qualitative measurements of the climbing assay. Top (green) indicates the flies present in the top 1 cm of the climbing tubes

(6.2-7.2 cm). Bottom (red) indicates the percentage of flies located in the bottom 1 cm of the tube (0-1 cm), and middle (blue) includes all flies that climbed between (1-6.2 cm). Significance was calculated using a cumulative link mixed model where the virus status and the time point were the fixed factors and experiments a random effect; false discovery rate contrasts were used for post-hoc analyses. For all the figure, *** $P < 0.001$, ** $P < 0.01$, * $P < 0.05$.
(TIF)

**S3 Fig. Bacteriome diversity analyses of pools containing at least 20 female flies per condition (uninfected or mono-infected). A)** Alpha diversity analyses on Chao1 and Shannon indexes reveal no significant differences in the species diversity of uninfected or persistently infected flies (pairwise Wilcoxon test on Chao1 and Shannon indexes, $P > 0.46$ for all comparisons). **B)** Principal component analysis of the Bray-Curtis dissimilarity matrix of the different studied fly populations at the OTU level. This analysis does not identify any specific differences in the bacterial community structure among the various fly lines (PERMANOVA on infection, $F = 1.57$, $P = 0.24$).
(TIF)

**S4 Fig. Principal Component Analysis of the transcriptional responese of the different samples sequenced.** These samples consist on pools of RNA coming from individual Drosophila females from the uninfecetd and infected stocks. Infection was confirmed through RT-qPCR. All samples in a pool (3-5) come from the same collection dpe and stock. On the Y axis of every graph, we indicate the second principal component (PC2) and the percentage of variance explained by this component between brackets. On the X axis, we indicate the first principal component (PC1) and the percentage of variance explained by this component between brackets. Black lines indicate the origin of each axis. The do represent our fly stocks throughout the different days post-eclosion, in green, we have the Uninfected stock, in burgundy, the Drosophila A virus persistent flies, in yellow, the Drosophila C virus persistently infected flies, in dark blue, we have the Bloomfield virus persistently infected stock, and in lavender the Nora virus persistently infected flies.
(TIF)

**S5 Fig. Volcano plots of the differentially expressed genes between the infected samples versus the uninfected flies.** Transcriptomic results indicate differentially expressed genes in our infected samples. Volcano plots showing the number of genes significantly up- (in red) or down- (in blue) regulated. In grey, we represent those genes displaying differential expression values lower than 1.5 log2 fold change and/or with p-adjusted values lower than 0.05. Gene names are indicated fot the top 10 genes in absolute Log2FoldChange values. These thresholds are indicated by dotted lines in the graphs. Each row presents the volcano plots of a specific fly stock over time (dpe), in the following order: DAV, DCV, Bloomfield, and Nora. Each infected sample is analysed in comparison to an uninfected control of the same age.
(TIF)

**S6 Fig. Top differentially expressed genes in the infected samples over the uninfected samples.** Persistent viral infection alters Drosophila melanogaster's transcriptomic profile in a virus-independent manner. For all heatmaps, on the X-axis, we indicate the infected stock analysed. **A)** From left to right: heatmap showing the top 200 differentially expressed genes in our samples in terms of absolute log2 fold change value at dpe1. Heatmap showing the top 20 DEGs at dpe1. **B)** From left to right: heatmap showing the top 200 differentially expressed genes in our samples in terms of absolute log2 fold change value at dpe12. Heatmap showing the top 20 DEGs at dpe12. For the heatmaps on the right, the top 20 gene names are indicated on the left of the graph
(TIF)

**S7 Fig. Common differentially expressed genes (DEGs) across infections and time. DEGs for each infected samples over their uninfected controls.** Venn diagrams of differentially expressed genes with differential expression values lower than 1.5 log2 fold change and/or with p-adjusted values lower than 0.05. **A)** DEGs for days 1, 12, and the day of

median survival for each infected individual stock. **B)** DEGs on the day of median survival of all persistently infected stocks. **C)** DEGs for each persistently infected stock over time.
(TIF)

## Acknowledgments

We thank all members of the Saleh lab for their insightful discussions. We are especially grateful to Vanesa Mongelli, whose early conversations sparked this project and who guided Mauro through his first steps in the *Drosophila* field.

## Author contributions

**Conceptualization:** Mauro Castelló-Sanjuán, Rubén González, Ottavia Romoli, Jared C. Nigg, Maria-Carla Saleh.

**Formal analysis:** Mauro Castelló-Sanjuán, Rubén González, Ottavia Romoli, Jared C. Nigg, Maria-Carla Saleh.

**Funding acquisition:** Jared C. Nigg, Maria-Carla Saleh.

**Investigation:** Mauro Castelló-Sanjuán, Rubén González, Ottavia Romoli, Hervé Blanc, Jared C. Nigg.

**Methodology:** Mauro Castelló-Sanjuán, Rubén González, Ottavia Romoli, Hervé Blanc, Jared C. Nigg.

**Validation:** Mauro Castelló-Sanjuán, Rubén González, Ottavia Romoli, Jared C. Nigg, Maria-Carla Saleh.

**Visualization:** Mauro Castelló-Sanjuán, Rubén González, Ottavia Romoli.

**Writing – original draft:** Mauro Castelló-Sanjuán, Rubén González, Ottavia Romoli, Jared C. Nigg, Maria-Carla Saleh.

**Writing – review & editing:** Mauro Castelló-Sanjuán, Rubén González, Ottavia Romoli, Hervé Blanc, Jared C. Nigg, Maria-Carla Saleh.

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
