## [Editor Report · Decision Letter 0]

10 Jun 2025

Dear Dr Saleh, dear Rubén,

Thank you for submitting your manuscript entitled "Persistent viral infections impact key biological traits in Drosophila melanogaster" for consideration as a Methods and Resources by PLOS Biology.

Your manuscript has now been evaluated by the PLOS Biology editorial staff, as well as by an academic editor with relevant expertise, and I am writing to let you know that we would like to send your submission out for external peer review.

Once your full submission is complete, your paper will undergo a series of checks in preparation for peer review. After your manuscript has passed the checks it will be sent out for review. To provide the metadata for your submission, please Login to Editorial Manager (https://www.editorialmanager.com/pbiology) within two working days, i.e. by Jun 12 2025 11:59PM.

Kind regards,

Melissa

Melissa Vazquez Hernandez, Ph.D.

Associate Editor

PLOS Biology

---

## [Decision Letter · Decision Letter 1]

15 Aug 2025

Dear Maria,

Thank you for your patience while your manuscript "Persistent viral infections impact key biological traits in Drosophila melanogaster" was peer-reviewed at PLOS Biology. It has now been evaluated by the PLOS Biology editors, an Academic Editor with relevant expertise, and by four independent reviewers. My sincere apologies for the extremely long peer-review process.

In light of the reviews, which you will find at the end of this email, we would like to invite you to revise the work to thoroughly address the reviewers' reports. As you will see below, majority of reviewers are positive about the relevance and novelty of the study, yet some concerns have raised during revision. Reviewer 1 mentions that he does not think this could be a Methods and Resources study and has some major comments that would require clarification or further discussion. Reviewer 2 thinks the study will be of general interest and a valuable resource and has some minor comments. Reviewer 3, is also positive saying "the study provides a novel and important resource", but has some concerns that would require modification of the claims or clarifications such as the method to stablish persistent viral infection and how it relates with more natural conditions, how virus evolution could affect the findings, and also asks to mention the limitations of the study. Reviewer 4 is also quite positive but suggests to quantify the total progeny production per population, as well as further discussion in some conclusions. The majority of the reviewers point out to the lack of access to the RNAseq data, which of course, would be something we require for publication. We agree with all reviewer concerns and would require some additional experimental revisions to address them, as we consider that this would strengthen the work.

IMPORTANT: given some of the comments, we would be open to change the format from "Methods and Resources" to a Short Report. However, this would require to shorten the number of main figures to 4. I am also happy to discuss with you any questions you may have during the revision process.

Given the extent of revision needed, we cannot make a decision about publication until we have seen the revised manuscript and your response to the reviewers' comments. Your revised manuscript is likely to be sent for further evaluation by all or a subset of the reviewers.

**IMPORTANT - SUBMITTING YOUR REVISION**

*Re-submission Checklist*

*Published Peer Review*

*PLOS Data Policy*

*Blot and Gel Data Policy*

Sincerely,

Melissa

Melissa Vazquez Hernandez, Ph.D.

Associate Editor

PLOS Biology

REVIEWERS' COMMENTS:

Reviewer #1:

The paper makes a thorough examination of the consequences of persistent viral infections, using 4 viruses in D. melanogaster, across a range of physiological traits and fitness proxies. The work seems of a high standard and takes a holistic approach, for example measuring viral load and host gene expression and multiple time points during the course of infection. The work is valuable for researchers working on Drosophila virus interactions, and examines infection dynamics and outcomes through this less studied (oral-fecal) route of infection.

The work is a natural progression of the authors previous work, however, it is somewhat incremental. As such the work is lacking in suitable novelty/originality to meet the PLoS Biology publication criteria, and is unlikely to have significance interest outside the field.

It is not clear why this is a methods and resources paper - whilst the authors have taken an interesting approach I do not feel this meets the PLoS Biology criteria for this type of article; it just seems to be a standard research paper. Methods are at the end and are brief and not at all novel - the major novelty is persistent infections but these are described in a previous paper by the group. In terms of resources, whilst it seems a well-executed and interesting piece of science and the results will be of interest to the field, it does not offer "resources of exceptional and general value" with "applicability and significant utility for the field and spur future research".

Major comments

For the study of viable offspring:

- Did you control the density of the populations the parental flies were taken from?

- Could some of the patterns (eg increased number of offspring in some viral infections) be due to reduced density in infected populations (and so larger flies from this reduced competition)?

- How was mortality accounted for? Eg should have seen ~25% death in DCV flies by day 10 post infection - were a greater number of offspring produced even if account for female mortality (e.g. could this be life-history trade off with terminal fecundity investment?). This is alluded to in the discussion, but you have the data to formally test, so this should be done. I would encourage some discussion/speculation on transgenerational effects also (e.g. do offspring from infected flies themselves have reduced fitness)

- For DCV in particular, where you see the greatest mean increase in total offspring (Fig 2b) you also see the greatest variation in this trait. When looking at figure 1 you see a bifurcation in infection outcomes with nearly a quarter of flies being uninfected by day 12, with a bimodal distribution for viral load also. As such I suspect that variation in offspring production may be due to some individuals being infected and some uninfected; although I think this makes the patterns all the more puzzling - this would benefit from some further analysis/thoughts.

RNAseq:

- Supp figure 2 should be in the main manuscript.

- Did you do RNA extractions on individual flies, check their infection status and then pool RNA (ie did you know whether all flies in your pools were infected/uninfected)? This needs clarifying and critical for interpreting DCV infection transcriptome data.

- Were you mapping reads only to D.mel or to your (and other) viruses as well (eg https://doi.org/10.3390/v15091849)? Helpful for above point and would be interesting to see if any other viruses are hanging around in your populations (an up-to-date list of know Drosophila viruses here https://obbard.bio.ed.ac.uk/data.html)

L476 - plot and test the trade-off (should be a negative correlation) between survival and reproductive output

Minor comments

Abstract - remove "challenge the assumption of the innocuousness of persistent viral infections for the host" - it is not a widely held assumption that persistent infections are benign.

First paragraph of the intro seems rather narrow in breadth.

- Firstly I would suggest incorporating some of the large scale sequencing studies for virus discovery eg https://doi.org/10.1016/j.cell.2024.09.027 and https://www.nature.com/articles/nature20167

- Secondly this would benefit from citing some studies other than the authors own on persistent viral infections to show a wide view on this type of infection

L479 - delete "to ensure population survival" - this reads as some slightly odd group selectionist argument which isn't needed.

L499 - 500 - cite a few more papers here to demonstrate the large amount of info on genetic variation in susceptibility to virus infection

More generally I felt there could be better scholarship and referencing of the literature, seemed to be picking just a couple of relevant examples rather than citing a broader range of relevant studies.

L510-511 add note saying lab microbiome probably not typical of wild flies

L588 - delete "comparative data"

Legends on supplementary figures need expanding as atm it is hard to tell (without ref back to main ms) which data they are referring too.

Data availability - an SRA number was provided but this data was not released. Data and code for non-sequencing data and analysis was also not available.

Reviewer #2:

In this manuscript, the authors investigated the impact of four persistent viral infections on several traits in Drosophila. All four viruses reduced the survival of flies but reached different load and prevalence in fly populations: DAV was present in all individuals, while DCV was cleared by most of the flies. The authors detected the effect of viruses on offspring production, sleep duration, and locomotor activity. Microbiome composition was not significantly affected by persistent viral infections, however total bacterial load significantly altered. Finally, the authors performed transcriptomic analysis of infected flies and different time-points and found that it is virus-specific and can persist even after clearence of the virus. Given a limited information on the impact of persistent infections on the host, this study will be of great general interest but also a valuable resource for Drosophila researchers. Overall, this is a well-performed and clearly-written work.

I have few suggestions to enhance the usability of data generated in this work.

1. Since this is a Resources paper, the authors should make their data as accessible and usable as possible. This is not the case for RNA-seq results. If someone wants to check if and how any gene changes expression under studied conditions, this is not possible to do with the information provided. The authors should provide all the transcriptomics results (tables with genes, their expression, and statistics) as supplementary files.

2. Volcano plots (sup fig 3). It would be useful to show at least for the most significant dots which genes stand behind them (like in fig 7b).

3. Can the authors please provide justification as to why they used whole fly samples and not guts for their RNAseq, given that infections are intestinal?

4. L604. Diet composition should be provided.

5. L607-610. It should be described how infections were established.

Reviewer #3:

In their paper, "Persistent viral infections impact key biological traits in Drosophila melanogaster", Castello-Sanjuan et al. show that persistent infection by four different viruses results in significant, virus-specific changes in viral titer, longevity, lifetime fecundity, locomotion, sleep time, microbial load in the digestive tract, and the whole body transcriptome. The experiments are generally well done, and the manuscript is well-written. Given its comprehensiveness and breadth, the study provides a novel and important resource to the field of host-microbe interactions and infectious disease. I commend the authors on the substantial amount of work that went into this study and the contribution they are making to the field. I have some concerns that should be addressed before publication, however, and they are outlined in detail below.

Major concerns:

1. Essential: The authors do not indicate whether the method used to establish persistent viral infection is representative of how flies would become persistently infected in a natural habitat. Moreover, the authors do not state whether persistent infection is common in natural populations for any of these viruses. Therefore, it is unclear whether this method to induce persistent infection is relevant to the natural ecology or evolution of any of these virus-host systems. This would not be an issue if the authors were solely trying to investigate the physiological effects of persistent infection on flies, but they attempt to connect these effects to the evolution of these viruses in natural populations. For these arguments to stand, they must make it clear whether their experimental setup in any way recapitulates naturally acquired persistent infection. Related to this question, assuming persistent infection is common in natural populations (which needs to be established and clearly explained by the authors), are the viral titers found in flies using this method similar to what would be found in a naturally infected fly with persistent infection?

2. How does evolution of the viral population potentially compromise the findings? Since the virus is being perpetually propagated among individuals from the same genetic background, is it possible it has adapted in a manner that is no longer representative of how wild viruses would impact the fitness of wild flies? What differences might be expected if the authors were to use recently isolated viruses? Again, the primary reason this is of concern is the stated implications of the findings for the evolution of these virus-host systems.

3. Essential: When measuring total offspring per female, the females were transferred to new food every day. Therefore, it seems likely to this reviewer that the persistence of viral infection would differ from that of the standard protocol used in this study (which was to transfer flies to new vials every two days). Can the authors speak to whether they have any data on persistence of viral infection in females during the fecundity experiment? If not, this should be presented as a potential caveat to the study.

4. Essential: The authors refer to Lactobacillus throughout the paper, but the most common OTU in their data set was assigned to Lactiplantibacillus (formerly Lactobacillus) which as far as I can tell, has been determined to be a separate genus from its former designation of Lactobacillus. See here for associated publications: https://lpsn.dsmz.de/genus/lactiplantibacillus Unless I am misunderstanding something, the authors should make this correction.

5. Essential: It is not clear to this reviewer why PCR amplification followed by sequencing failed to yield any sequences in Bloomfield infected flies, but qPCR was successful. Given that both approaches have an amplification step, why was it successful for one but not the other? The authors don't mention negative controls in the methods for their microbiota sequencing or qPCR. I assume they were used but this should be clearly stated. For qPCR, please clarify how the negative controls were used to distinguish true signal from contamination. For example, indicating that the negative controls yielded no signal or that the signal obtained from the negative controls was sufficiently weaker than that in Bloomfield flies. Otherwise, because of the ubiquity of contaminating 16S DNA in any laboratory, it is impossible to know if the Bloomfield signal is real or just environmental contamination.

6. Essential: In general, pooling individuals for microbiota sequencing in species with environmentally acquired microbiota is not ideal, because there can be substantial inter-individual variation and a single individual with a high amount of one OTU can skew the entire pool, resulting in incorrect conclusions. I know pooling is common practice in Drosophila, perhaps because of limited inter-individual variation in microbiota composition, but I suggest moving away from this practice whenever time and funding allows. I note that pools are not used by the authors in any other assay (except for transcriptomics), therefore I assume they recognize the importance of accounting for inter-individual variation. I do not suggest the authors need to collect more data for this study, but including a comment and supporting citation(s) in the methods justifying the approach of pooling or discussing its limitations is called for.

7. Essential: At what taxonomic level were the microbiota diversity analyses performed? This should be indicated in the methods. Also, what variables were included in the PERMANOVA? To what variable does the p-value provided refer?

Minor concerns:

Line 717 OTU, not OUT.

The authors refer to OTU classification in the methods but bacterial species throughout the study. Was species level ID obtained for the vast majority of OTUs? If not, OTU would probably be a more appropriate term to use.

Figure 5, the legend indicates that "Shape fill indicates the replicate" but there are only two shape fills and each is used for three data points. Therefore, I am confused as to what shape fill is indicating and the general replication structure of the qPCR data.

Line 529: The authors refer to "The systemic impact…" but don't explain what they are referring to here? Is the assumption that the impact of infection on locomotion is a systemic impact because it's outside the gut? Please make the connection more clear, as it was confusing the first time I read through it.

Supplementary Figure 1: Why does Uninfected have error bars but no other treatment? Please make this uniform.

Reviewer #4:

This manuscript presents a detailed study of the effects of four viruses causing persistent infections in the fitness, behavior, transcriptomic patterns and microbiome of their host Drosophila melanogaster. Although persistent infections are not uncommon, most studies focused on acute viruses. Thus, this study is scientifically relevant and timely as it tackles an important biological question, yet only partially understood. The manuscript is well written and easy to follow. The data presentation helps the reader navigate through the breath of experiments done and support the conclusions of the work. Thus, I think this is a neat piece of science and I only have some minor comments:

- Figure 1c. I´m not sure of understanding the pie diagrams. According to panel 1b, all viruses induce certain level of mortality from 20 dpe on, or even earlier. If so, the number of flies analyzed in panel 1c should not be constant over time as certain mortality is expected and prevalence should be calculated over surviving individuals. Were all flies surviving until 28 (or even 52) dpe? If so, how does this result fit with those on panel 1b?

- The authors quantify effects on host fitness as production of viable progeny per female. This is clearly a good proxy and I am not criticizing it. However, in various parts of the manuscript the authors mention that virus-induced changes in host fitness have consequences for the host population. To address this question, I think that, rather that descendants per female, the relevant parameter is the total progeny production per population (per tube or per treatment in this case). This measure should take into account not only the total number of descendants produced per female, but also how many females survived to produce progeny and the prevalence of the virus in the population (i.e., the total number of descendants in the population will be the sum of that produced by infected and non-infected individuals. Did the authors quantify this parameter? It should be quite straightforward as all the necessary information is already available.

- Lines 476-485. The authors invoke a survival-fitness trade-off to explain their results. However, some of the their results might not be in line with this conclusion. First, all hosts (infected or not) produced progeny in the first 12-15 dpe. Except for DCV, virus-induced mortality is negligible at this point. Thus, virus infection is apparently affecting survival after progeny is produced. Second, the authors discus that higher progeny production comes at the cost of lower immunity, which would explain higher mortality of infected individuals. If so, I would expect those viruses inducing higher progeny production to multiply to higher levels in the host as immunity would be weaker. Nevertheless, DCV and Bloomfield, which induce the highest increases in progeny production are also those with lower virus multiplication levels, suggesting an active immune response. An alternative explanation for these results is that the virus adapted to remain at low levels in order to allow progeny production such that its chances for transmission are enhanced (i.e., the population of susceptible hosts is larger). I would like to know the authors opinion about this possibility.

- Lines 488-490. If viruses induce accelerated aging, would not shorter reproductive periods be expected?

- A curiosity. I came across recently over several papers showing that genes traditionally considered as good normalizers of virus titer are indeed affected by the virus presence. Did the authors check that Rp49 expression was not affected by virus infection?

- Finally, I detected some typos: (i) Line 41, species name should be in italics. (ii) Fig 2a, please review the x axis, values 14 and 15 overlap. (iii) Line 522, there is a dot before the reference.

---

## [Editor Report · Decision Letter 2]

19 Sep 2025

Dear Dr Saleh,

Thank you for your patience while we considered your revised manuscript entitled "Persistent viral infections impact key biological traits in Drosophila melanogaster" for publication as a Short Report at PLOS Biology. This revised version of your manuscript has been evaluated by the PLOS Biology editors and the Academic Editor.

Based on our Academic Editor's assessment of your revision, we are likely to accept this manuscript for publication, provided you satisfactorily address the data and other policy-related requests stated below my signature.

We expect to receive your revised manuscript within two weeks.

*Published Peer Review History*

*Press*

Sincerely,

Ines

--

Ines Alvarez-Garcia, PhD

Senior Editor

PLOS Biology

on behalf of

Melissa Vazquez Hernandez, Ph.D.

Associate Editor

PLOS Biology

Fig. 1B-G; Fig. 2A-C; Fig. 3A, B; Fig. 4A-D; Fig. S1; Fig. S2A, B; Fig. S3A, B; Fig. S4; Fig. S5; Fig. S6A, B and Fig. S7

Please also ensure that figure legends in your manuscript include information ON WHERE THE UNDERLYING DATA CAN BE FOUND, and ensure your supplemental data file/s has a legend.

CODE POLICY

---

## [Editor Report · Decision Letter 3]

24 Sep 2025

Dear Carla,

Thank you for the submission of your revised Short Reports "Persistent viral infections impact key biological traits in Drosophila melanogaster" for publication in PLOS Biology. On behalf of my colleagues and the Academic Editor, Harmit Malik, I am pleased to say that we can in principle accept your manuscript for publication, provided you address any remaining formatting and reporting issues. These will be detailed in an email you should receive within 2-3 business days from our colleagues in the journal operations team; no action is required from you until then. Please note that we will not be able to formally accept your manuscript and schedule it for publication until you have completed any requested changes.

PRESS

Sincerely, 

Melissa

Melissa Vazquez Hernandez, Ph.D., Ph.D.

Associate Editor

PLOS Biology
